# Here Puppy, Chew on This: Short-Term Provision of Toys Does Not Improve Welfare in Companion Dogs

**DOI:** 10.3390/ani13213340

**Published:** 2023-10-27

**Authors:** Kelly Chan, Carol Arellano, Alexandra Horowitz

**Affiliations:** Dog Cognition Lab, Department of Psychology, Barnard College, New York, NY 10027, USA; cgarellano2@gmail.com (C.A.); ahorowit@barnard.edu (A.H.)

**Keywords:** canine welfare, dog toys, enrichment toys, environmental enrichment, canine cognition

## Abstract

**Simple Summary:**

Burgeoning interest in the emotional experiences and cognitive abilities of dogs has led to novel methods to measure and improve canine welfare. Dog toys, in particular, are commonly thought to benefit dogs’ mental and physical health. In this study, we examined the effect of introducing new toys to companion dogs. Our results showed a slight improvement in some positive welfare measures in dogs who received more complex toys when compared to dogs who received less complex toys and those who received no toys. However, this difference was not statistically significant and we found no differences in subjects’ eating rate or activity levels. While research has found that toys may improve the welfare of kenneled dog populations (shelter- and laboratory-housed dogs), our findings did not find the same to hold with pet dog populations. We discuss the relevance of the study population and the welfare measures used in attempts to study “enrichment” in animals.

**Abstract:**

Retail dog toys are often provided to companion animals to provide cognitive and physical stimu- lation and improve the animals’ quality of life. These toys, sometimes known as “enrichment toys”, have been shown to play a role in increasing appetite and activity levels and decreasing undesirable behaviors (e.g., barking, self-isolating behaviors) in some domestic dog (*Canis familiaris*) populations. In this study, we evaluate the effect of toys on appetite, activity levels, and positive affective states as measures of well- being in companion dogs. Behaviors were compared before and after regular interactions with different types of toys over twelve days. We found that provisioning companion dogs with toys did not significantly alter their activity level, rate of food consumption, or cognitive bias. While dogs who received more complex toys showed a slightly improved cognitive bias, there were no significant differences in behaviors be- tween the subjects who received “less complex” toys (e.g., a bone, ball) and “more complex” toys (e.g., puzzle toys). We conclude with reflections on the relevance of our subject population to the result seen, and on the different forms of the cognitive bias test.

## 1. Introduction

Over the last several decades, the concept of animal welfare has evolved to reflect the changes in our knowledge of animal behavior. Many early notions of animal welfare derived from attitudes about the basic needs of animals [1,2] and primarily focused on suffering or stress as measurements of welfare [3,4,5,6]. More recently, the focus in animal welfare has included an interest in understanding animals’ cognitive abilities and subjective experiences and working toward introducing positive experiences [2,7,8]. Researchers rely on physiological, behavioral, and cognitive measures to investigate how animals, especially captive animals, cope with their given environments [7,9,10,11]. However, these measures can be susceptible to differing interpretations, and, thus, the implications for welfare are non-obvious. Watanabe [12] argues that animal welfare is a social construct that will continue to develop as we learn more, especially about how animals’ cognitive and sensory abilities contribute to their well-being.

The field of animal cognition has historically focused on non-human primates and captive mammals; this restriction has been loosened in the last two decades to include research on domestic animals from pets to farm animals. This research has revealed that complex social and cognitive skills are not restricted to primates. For instance, the domestic dog (*Canis familiaris*) has been shown to display higher-order object permanence [13,14], episodic-like memory [15,16], and comprehension of human social cues [17,18,19]. Recently, research on internal affective states and cognitive processes such as preference and motivation has provided insight into potential indicators of welfare [20,21,22]. The presentation of novel stimuli and observations of the animals’ subsequent behavioral response is commonly used to investigate subjective experiences in animals [23].

For captive animals, environmental enrichment, or the provision of stimuli (e.g., physical, social, feeding) in and modifications to the environment, is said to improve animals’ quality of life [24,25,26]. For dogs, specially designed retail toys are common sources of stimulation. A recent survey-based study on companion dogs found many people believe feeding enrichment toys provide mental stimulation and prevent boredom in dogs [27]. Dog toys are widely touted to improve dogs’ well-being through facilitating natural exploratory behavior and mitigating stress-related, maladaptive behaviors such as over-grooming and destruction of furniture [28,29,30,31]. In fact, Flint et al. [32] supplied kenneled dogs with a variety of food-based enrichment and found long-lasting chews to be the most effective as measured by the dogs’ positive affective state and time spent engaging with the toys. It may be that the act of chewing itself is rewarding as it engages different senses and promotes oral health in dogs [31,33].

These claims were borne out of research in limited and variable contexts. For instance, Hubrecht [28] found that shelter dogs who were given simple chew toys exhibited increased activity and decreased undesirable behaviors such as destruction of furniture. Herron et al. [34] provided shelter dogs with behavioral training and toys, both involving food rewards; they also found a decrease in both undesirable behaviors such as barking and jumping when greeting, and in general activity (which was deemed desirable in the shelter setting). In another study, the provision of food within toys (or feeding enrichment) to laboratory-housed dogs also found a significant decrease in time spent inactive, as well as an increase in measures of locomotion and appetitive behaviors [35].

The generalizability of these limited findings is not clear. Most environmental and feeding enrichment research has studied dogs housed in kennel, shelter, or laboratory conditions. Moreover, there is considerable variability in experimental design (duration of interaction, types of toys, limitations of the shelter environment). Only one study [27] has investigated the effects of toys on the welfare of companion dogs living in households, however, only survey-reported data was collected.

In the current study, we aimed to gauge whether provisioned toys are in fact enriching for companion dogs. In particular, we looked at the effect of introducing such toys for two weeks on external (subjects’ appetite, activity levels), and internal (cognitive bias) measures of well-being. As there is an abundance of toys available to consumers, we also aimed to explore whether the introduction of toys with more complex form or function than subjects were accustomed to would have a salutary effect.

We collected data on the length of subjects’ daily eating time, their activity rate, and their cognitive bias before and after the introduction of the specified toys. Following given instructions, daily food consumption information was provided by the subjects’ persons. Activity rate was gauged by FitBark 2 dog activity monitors (Kansas City, MO, USA) using their proprietary activity measure. Subject cognitive bias was gauged through the cognitive bias test. First introduced by Harding et al. [36], the cognitive bias test is commonly used to observe how animals perceive and choose to interact with ambiguous stimuli and infer how internal states influence such behavioral responses [37]. In particular, it is used as an indication of subject welfare. In the test, subjects are first presented with both rewarding and non-rewarding stimuli in designated locations followed by presentation of an “ambiguous” stimulus in a novel position directly between those locations. The stimuli used have varied across species studies, including light intensity [38], but more usually food provisioning. The cognitive bias test protocol is robust across species, including honeybees [39], rhesus macaques [40], starlings [41], dogs [42,43], and New Caledonian crows [44].

We hypothesized that using provisioned toys that are more complex than available toys will improve dogs’ welfare, as measured by their cognitive bias, appetite, and activity level. If toys are enriching, we expect a decreased latency for dogs to approach an ambiguous stimulus, increased activity levels, and increased duration of food consumption—which may be considered a measure of appetite—after short-term engagement (12 days, 5 min daily) with the provided toys. While this study was designed to gauge the effect of toys on companion dogs’ welfare, we were also interested in exploring the usefulness and practicality of these measures of welfare in this subject population. Furthermore, given that there are different ways to test cognitive bias, we discuss the implications of our work on future cognitive bias studies, as well as the effect of person-induced bias in research.

## 2. Materials and Methods

### 2.1. Subjects

Subject dogs and their people were recruited through a Barnard Dog Cognition Lab database as well as in-person and virtual flyers. Only dogs who were over six months old at the time of recruitment, who had lived with their current families for at least three months, and who were food-motivated were invited to participate. There were no other restrictions on dog breed or age.

Forty-seven domestic dogs (25M, 22F) between the ages of 6 months and 11.9 years old (M = 4.47 ± 2.88 years) participated in our study (Table 1). On average, subjects had lived with their current families for 3.5 years. Thirty dogs were mixed-breed and thirteen dogs were purebred (Australian cattle dog (1), English Springer Spaniel (1), Pug (1), Brussels Griffon (1), Newfoundland (1), Shih Tzu (1), Cavalier King Charles Spaniel (1), Fox terrier (1), Border Collie (1), Shetland Sheepdog (1), Affenpinscher (1), Havanese (1), Wheaten Terrier (1)). All but one dog had toys accessible to them at all times and 25% of dogs had food accessible to them at all times. Approximately 21% of dogs were reported on entry surveys to exhibit separation-related behaviors.

### 2.2. Procedure

A questionnaire completed by the subject dogs’ families provided us with details of each dog’s biographical information, including typical feeding and playing schedules, experience with dog toys, and duration of time spent alone at home. Subjects were randomly assigned to an experimental group (N = 32) or a control group (N = 15). Seven subjects did not complete the study and were not included in the analyses.

Experimental trials were conducted in the Dog Cognition Lab on the campus of Barnard College (New York, NY, USA) from February 2022 to December 2022. The testing space was a room measuring 3.53 × 3.35 m and was absent of furniture. The average temperature of the room during the trials was 23.56 C and the average humidity was 42.7%. Lorex cameras (LBV2531W model; Markham, ON, Canada) were used to record subjects’ in-trial behaviors (Figure 1).

Subjects visited the lab twice, each time to participate in a cognitive bias test (described below) before and after the intervention. Prior to the first visit, baseline activity levels and appetite were collected from subjects’ unchanged routines by their persons. To measure appetite, a timer was started when food was made available to the dog and was stopped when the dog moved away from the bowl for more than five minutes without returning. Data on activity levels were collected by a FitBark 2 activity monitor, which contains a 3D accelerometer.

For twelve days between the visits, people in the experimental group were given instructions on how their dogs should engage with the toys provided (see Experimental Conditions). Control group subjects were asked to socialize with their dogs for a similar length of time, while giving food rewards. In cases (N = 13) in which the time between Lab visits was more than two weeks, the intervention behavior and data collection were continued until the time of their second visit (Table 2).

#### 2.2.1. Cognitive Bias Test

Procedures for the cognitive bias test procedures were drawn from Mendl et al. [42] and Duranton and Horowitz [43]. For each trial, the dog’s person was seated on a chair, with their dog sitting in front of them between their legs (see Figure 1), facing the experimenter 2 m away who presented the stimulus (bowl) in a specified location within a stimulus array, dependent on the type of trial. Another experimenter stood 1.5 m from the subject, removed from the experimental stage, and used a timer to record subjects’ latency to approach the stimuli in all trials. For every trial, subjects were allowed up to 30 s to freely investigate the stimuli.

Training

For each training trial, the experimenter presented the stimulus—a bowl baited or non-baited with one piece of food (PureBites beef liver treats) depending on the condition—at a location 0.5 m to the left or right of a center point. When the bowl was placed on the “positive” side (P-trial), it contained a food reward and when the bowl was placed on the “negative” side (N-trial), it was empty. The location of the positive and negative sides was randomly assigned, for each subject and visit, to be at the right or left (of the experimenter) locations.

Each subject received two P-trials and two N-trials before receiving trials in a pseudo-randomly selected order. Subjects did not receive more than two of the same trial types consecutively and received a minimum of 14 training trials before being considered for test trials. Subjects were considered “trained”, or having discriminated the positive and negative locations, when their lowest latency to approach baited bowls (P-trials) was higher than the highest latency to approach non-baited bowls (N-trials) over the previous four consecutive trials.

Test

Subjects received up to six test trials with an “ambiguous” bowl (A-trials). In each, an unbaited bowl was placed between the P and N bowl locations. Two training trials (P and N) were conducted after three test trials to mitigate the effects of receiving consecutive unbaited bowls at the ambiguous location [42]. All subjects received a final trial (P-empty trial) in which an unbaited bowl was placed at the positive location. Thus, the testing phase included a total of nine trials.

For both training and testing phases, subjects moved onto the next trial if they did not visit the bowl within 30 s. If they did not approach the bowl over three consecutive trials, their participation in the study concluded. This was the case for one subject and their data were excluded from analyses. Refer to the Appendix A for more details on the procedure for the cognitive bias test.

#### 2.2.2. Experimental Conditions

##### Experimental Group

Within this group (N = 30), our protocol intended that subjects were assigned to subgroups (“levels”) receiving different toys based on the survey response of the subjects’ person on their dog’s favorite toy. Levels of toys varied based on the increasing complexity of engagement required by the dog (Table 3). “Level 1” toys (Nylabone DuraChew Original Bone (Neptune City, NJ, USA), Max and Neo Furry Pals Water Bottle Toys (without inserts, Scottsdale, AZ, USA), ChuckIt! Ultra Ball (Syracuse, NY, USA)) were chewing and mouthing toys, which typically engage one or two body parts (mouth, paw) and a single sensory modality (touch). “Level 2” toys (LickiMat (Sydney, Australia), Kong Original (Golden, CO, USA), Starmark Treat Crunching Barbell (Hutto, TX, USA)) add a food reward component, encouraging additional exploratory behaviors such as nosing and licking, and involving the taste sense. “Level 3” toys (Nina Ottosson Puzzles; Treat Maze, Brick, Tornado (Karlskoga, Sweden)) engage those sensory modalities and also require active investigation to obtain food rewards.

Subjects were grouped in toy levels based on the survey-reported “favorite toys” or their presumed toy preference, which was considered their baseline. Due to experimental error, subjects were pseudo-randomly assigned to different toy levels. Thirteen subjects were assigned toys that were above their favorite toy’s level and seventeen subjects were assigned toys of the same or below their favorite toy’s level.

Over twelve consecutive days after their first visit to the lab, subjects’ persons were asked to provide a toy for five minutes daily, rotating a new toy every four days to avoid habituation [26,30,45], and to encourage interest, but not force interactions, with the toys. To encourage compliance, they were given a distraction task of counting the number of times the dog stops and resumes play with the toy on a datasheet.

##### Control Group

To balance the groups for food reward and persons’ attention, persons from the control group (N = 15) were instructed to socialize with their dogs while giving them treats for five minutes daily for twelve days or up until their second visit.

### 2.3. Coding

#### 2.3.1. Cognitive Bias Test

Subject latency to approach the stimuli in the cognitive bias test was coded in frames per second (fps) by two researchers using the software BARC (New York, NY, USA) [46]. Latency was measured as the time between the release of the dog from the starting position and when the subject’s head dipped into the bowl or their nose touched the rim of the bowl [42] or approached even without touching the bowl. The dog was considered to have been released when the person removed their hand from their dog’s collar or when the leash was dropped onto the floor. Any irregularities in the running of the trial or unusual dog behavior were noted by the coder. All behavioral coding was performed by coder K.C.; a second coder, R.W., who was blind to the subjects’ groups, coded 20% of the videos. To ensure inter-coder reliability, a Spearman’s rank correlation was computed and there was a positive correlation between the two coders (*r*(16) = 0.97, *p* < 0.001).

#### 2.3.2. Food Consumption

Data on the duration (in minutes and seconds) of daily meal consumption were collected for all subjects for the entirety of the subjects’ participation. If subjects were provided with more than one meal each day, data were collected for the same type of meal (breakfast or dinner).

#### 2.3.3. Activity Level

BarkPoints generated by the FitBark 2 activity monitor were automatically uploaded and synced onto an online platform (web.fitbark.com, accessed on 1 August 2023). Data on the daily number of BarkPoints were collected at the conclusion of all subjects’ participation in the study.

### 2.4. Behavioral Analysis

We performed both between- and within-group comparisons, looking at all behavioral measures both pre- and post-intervention. We initially intended to compare the experimental and control groups as our primary comparison, but due to experimental error in toy assignment, post-hoc analyses were conducted on the more appropriate comparison: between the experimental group subjects who received higher-level toys (“higher-level subjects”; N = 15) and lower-level toys (“lower-level subjects”; N = 17). We also compared control and experimental groups (higher-level toy subjects only) at both timepoints.

Changes in cognitive bias, appetite, and activity level were compared separately. All analyses were conducted first including all relevant subjects, then excluding subjects due to differences in experimental timelines or insufficient data.

#### 2.4.1. Cognitive Bias Test Protocol

To account for individual differences in subjects’ speed to approach, the difference in latencies in the A-trials pre- and post-intervention was calculated. They were then averaged and converted from fps to seconds for analyses. Latencies in the training trials (P, N) and the unbaited P-bowl (P-empty) were coded as well. Trials in which subjects did not approach within the maximum observation period of 30 s were excluded from analysis. To identify differences in subject participation in the testing phase, the number of A-trials subjects participated in was compared pre- and post-intervention.

Primary analyses of cognitive bias compared the raw or unadjusted averages of subjects’ Atrial latencies. Considering the possibility of subject habituation to the stimuli and adjustments to latency data, we also performed post-hoc analyses comparing adjusted averaged A-trial latencies, using a formula derived Mendl et al. [42], for the first A-trial from pre- and post-intervention. More details on the procedure and behavioral analyses of these comparisons is provided in Appendix A.

##### Outliers and Exclusions

All latency data from both pre- and post-cognitive bias tests were pooled to determine outlier values beyond the calculated upper limits of 2.5 standard deviations from the mean. Five subjects’ data were determined to be outliers. These data points were presumed to be due to extraneous environmental variables (such as distraction or human influence) or experimental error and were excluded from all analyses. In addition, two subjects from the higher-level experimental group did not participate in a post-cognitive bias test; their data were excluded from analyses as well.

#### 2.4.2. Food Consumption

Reported feeding time prior to the first Lab visit was used as baseline data. Post-intervention data were collected in the three days before the subjects’ second visit. Raw data were converted to minutes for analysis. Three subjects were excluded from between-group analyses due to incomplete data at either time points; five subjects were excluded from all analyses due to missing data at both time points. Subjects’ average duration of food consumption pre- and post-intervention were compared.

#### 2.4.3. Activity Levels

Subjects’ baseline activity was set by the BarkPoints (FitBark 2) average reported for three days prior to the first visit. Post-intervention data were collected in the three days before the subjects’ second visit. Three subjects were excluded from all analyses of activity levels due to missing data and one subject was excluded from between-group comparisons due to insufficient data. Subjects’ average BarkPoints pre- and post-intervention were compared.

### 2.5. Statistical Analysis

Graphical and statistical (Shapiro–Wilk test) methods indicated that our data were not normally distributed; as a result, non-parametric statistics were used for all analyses. Wilcoxon tests were conducted for within- and between-groups comparisons and the Pratt method was applied to address ties in the data; an asymptotic distribution was assumed [47]. Effect sizes and confidence intervals were conducted using the Glass rank biserial coefficient [48,49]. Analysis of survey-reported data on behavior were conducted using Wilcoxon and Kruskal–Wallis tests. All analyses were run using R version 4.3.0 (Vienna, Austria) [50].

## 3. Results

Subjects took a mean of 19 (±5; range: 14–38) trials to meet the training criterion and move to the testing phase. In the second visit, subjects participated in a mean of 18 (± 5; range, 14–30) training trials before moving onto the testing phase. For all subjects, any non-approaches to the ambiguous bowls were excluded from subjects’ average latencies. Thus, subjects participated in varying numbers of A-trials. Subjects approached A-trials significantly more often pre- (M = 5.51 ± 0.73 trials) than post-intervention (M = 4.71 ± 1.47 trials; Wilcoxon signed rank: Z = 3.82, *p* < 0.001).

To test the possibility that subject behavior was influenced by odor cues, we conducted Wilcoxon tests comparing latency to the empty bowl presented in test trials (P-empty) with latency to the baited bowl (P-test) (N = 44; two subjects, one at each time point, did not approach the P-empty bowl) [42]. There was no difference in latency to approach the P-empty bowl (M = 3.23 ± 2.89 s) and the bowl in the P-trial (M = 2.52 ± 1.43 s) for all subjects pre-intervention (Wilcoxon Mann–Whitney: Z = −1.23, *p* = 0.220) or post-intervention (P-empty, M = 3.01 ± 2.60 s; P-test, M = 2.62 ± 1.65 s) (Wilcoxon Mann–Whitney: W = 976, *p* = 0.950). This suggests subject behavior was not influenced by odor cues.

Age, sex, and breed (mixed or purebred) were not correlated with latencies to approach the A-bowl (averaged and first trial), appetite, or activity levels. Lifestyle differences, such as the daily amount of time spent interacting with their persons and exercising, the presence of separation anxiety-related behaviors, and the availability of food, were also not correlated with subjects’ behaviors.

### 3.1. Higher-Level Toy vs. Lower-Level Toy Subjects

#### 3.1.1. Cognitive Bias Test

Based on averaged A-trial latencies, subjects who received the higher-level toy (N = 12; unadjusted: M = −0.23 ± 1.44 s; adjusted: M = −469.16 ± 1546.47 s) approached the ambiguous stimulus faster than subjects who received lower-level toys (N = 15; unadjusted: M = 0.24 ± 2.61 s; adjusted: M = 11847.71 ± 46500.22 s). However, the difference in unadjusted average latencies was not significant (Wilcoxon Mann–Whitney: W = 93, *p* = 0.905; *rrb* = 0.03, 95%, CI = [−0.39–0.45]) after the intervention. Nor was there a significant difference between higher- and lower-level subjects after adjusting their average A-trial latencies (Wilcoxon Mann–Whitney: W = 97, *p* = 0.755). The same result was observed between higher-level dogs (N = 10) and lower-level dogs (N = 12) after omitting five subjects due to experimental timeline differences (Wilcoxon Mann–Whitney: W = 57, *p* = 0.872).

To examine the possibility that repeated exposure to A-trials may have introduced learning effects in the subjects, we conducted additional analyses using subject latencies in only the first A-trial. Subjects’ first A-trial approach was 2.57 ± 2.24 s on average. There were no significant differences in change in first A-trial latency in higher-level subjects (N = 13) and lower-level sub-jects (N = 16) from pre- to post-intervention (Wilcoxon Mann–Whitney: W = 84, *p* = 0.398).

#### 3.1.2. Food Consumption

Seven subjects (five higher-level, two lower-level) were omitted from all analyses on appetite due to incomplete data. There was no significant difference in the average time spent eating in subjects who received higher-level toys (N = 10; M = 0.26 ± 1.24 min) and lower-level toys (N = 15; M = −0.66 ± 1.94 min) after participating in enrichment sessions (Wilcoxon Mann–Whitney: Z = −0.10, *p* = 0.318). The introduction of higher-level toys did not have an effect on the duration of food consumption rate.

#### 3.1.3. Activity Levels

Two subjects from the higher-level toy group were omitted from activity-level analyses due to incomplete data. While all subjects showed a decrease in activity level (BarkPoints) after the intervention, there was no difference in the average activity level of higher-level subjects (N = 13; M = −763.72 ± 2500.29) and lower-level subjects (N = 17; M = −773.2 ± 1687.344) from pre- to post-intervention (Wilcoxon Mann–Whitney: W = 97, *p* = 0.592).

### 3.2. Higher-Level vs. Control Subjects

Control subjects differed from lower-level subjects in that they did not receive any toys and, instead, received regular socialization and food rewards from their humans. The control condition was designed to contrast with the experimental condition; given the error in toy assignment, we can instead compare the behavior of the control subjects (N = 15) and the experimental subjects who received higher-level toys (N = 12; three dogs were excluded due to incomplete data), as the lower-level subjects may have accidentally been assigned toys which were less complex than their typical toy.

#### 3.2.1. Cognitive Bias Test

Subjects who received higher-level toys approached the ambiguous stimuli faster (M = −0.23 ± 1.44 s) after the intervention than dogs who did not receive toys (M = 0.51 ± 3.71 s); however, this difference was not significant (higher-level N = 12; control N = 15; Wilcoxon Mann-Whitney: W = 109, *p* = 0.373). Similarly, after omitting six subjects who deviated from the 12-day research protocol, there is still no significant difference in the average latency to approach in the A-trials between higher level subjects (N = 10; M = −0.20 ± 1.58 s) and control dogs (N = 11; M = 1.35 ± 3.46 s) after the intervention (Wilcoxon Mann-Whitney: W = 70, *p* = 0.314).

Higher-level subjects appeared be quicker to approach the A-bowl after the first trial (first: M = 0.24 ± 0.40 s; average: M = −0.23 ± 1.44 s), whereas control subjects (first: M = −0.50 ± 1.18 s; average: M = 0.51 ± 3.71 s) appeared slow in their approach after their first A-trial. Although the toys did not have an effect on promoting a more positive cognitive bias in companion dogs, these results suggest that idiosyncrasies of individual dogs may contribute importantly to the observed changes in behavior.

#### 3.2.2. Food Consumption

Six subjects (five higher-level, one control) were omitted from all analyses on appetite due to incomplete data. There was no significant difference, from pre- to post-intervention, in the average time spent eating between higher-level subjects (N = 10; M = 0.26 ± 1.24 min) and control subjects (N = 14; M = −0.22 ± 1.57 min; Wilcoxon Mann–Whitney: Z = −0.64, *p* = 0.519).

#### 3.2.3. Activity Levels

Four subjects (two higher-level, two control) were omitted from activity-level analyses due to incomplete data. There was no significant change when comparing the activity level of higher-level dogs (N = 13; M = −763.72 ± 2500.29) and control dogs (N = 13; M = 203.73 ± 1984.77) after the intervention (Wilcoxon Mann–Whitney: W = 95, *p* = 0.614).

## 4. Discussion

This study aimed to investigate the effect of dog toys on internal and external measures of welfare in companion dogs. In addition, we aimed to attach toy level to efficacy as an enrichment device. In our study, the provision of toys in the intervention induced no significant changes in companion dogs among any behavioral measures of interest. Dogs who received higher-level toys generally approached the ambiguous stimuli faster, after 12 days with the toys, than dogs who did not receive higher-level toys (Table 4 and Table 5). The experimental group also showed increased appetite and lower activity levels after participating in the intervention. However, these differences were not significant; thus, we cannot make any strong conclusion about the contribution of the toy intervention to the result. Nor was there any significant decrease in welfare among most of these measures after the experimental intervention.

While our results are non-significant, they are worth reporting since they differ from previous studies looking at the effect of introducing toys to subject dog populations. Furthermore, our study provides an occasion to reflect not only on the relevance of the subject population studied, but also on the differing measurements of positive welfare, especially the different analytic approaches used in the cognitive bias test. Finally, we highlight some methodological challenges we faced in this research. Below, we discuss all of these elements.

### 4.1. Subject Population

Our results do not align with the results of empirical research on kenneled dog populations, which find some improvements in appetite and activity for dogs provisioned with toys. The discrepancy may be explained by the baseline environmental conditions for these subjects. For various reasons related to the need to keep dogs separated, kenneled subjects experience small enclosure spaces, few to no opportunities to socialize with conspecifics and humans, limited access to outdoors, and a lack of furniture [28,30,51,52,53,54]. Kenneled dogs have also been observed to spend most of their time inactive [28,35,55]. As a result of these conditions, it is safe to assume that companion dogs, living in human homes, are generally exposed to more complex environments and activities than kenneled dogs. Providing toys to dogs who already have regular access to different toys as well as the opportunity to interact with humans and other dogs, may provoke, at best, a smaller change in the typical measures of welfare, such as the ones used in this study (appetite, activity, and cognitive bias). In other words, their environment is already “enriched”, per the measures used in previous work.

### 4.2. Measuring Welfare: The Cognitive Bias Test

Across past instances of the application of the cognitive bias test, different measures of the A-trial latencies have been used. For the most part, we did not find any changes to the significance results when looking at the A-trial behavior in three different ways: as average A-trial latencies; as first A-trial latencies; and as adjusted A-trial latencies (in two comparisons of first Atrial latencies, the higher-level subjects were significantly slower than the control subjects). However, a closer examination of the data indicated subtle differences that may have been more meaningful with, say, a larger subject group. For instance, higher-level subjects (M = 0.24 ± 0.71 s) took longer to approach the ambiguous stimuli in the first A-trial than lower-level subjects (M = 0.01 ± 0.40 s) from pre- to post-intervention (Figure 2). Similar results were found on omitting the five subjects omitted in the main analysis for their irregular timelines. This relationship is in contrast to that found when comparing average A-trial latencies between groups, and prompts the question of which way of measuring latency is more accurate when trying to capture cognitive bias changes.

Other variations in cognitive bias test procedures include the number of ambiguous probes and their locations, training criteria, and probe visibility. The salience of these separate variables in the results found has not been examined. Relatedly, in our study, an unexamined measure—the number of trials that subjects participated in—was noticeably different across subject groups. Half of all subjects displayed a significant drop-off in participation in A-trials from their first to second visits; the other half participated in the same number of A-trials. (All subjects participated in an average of more than four A-trials at both time points). Might this measure, in conjunction with latency data, be useful in providing additional context to changes in cognitive bias?

In theory, longitudinal measures need to be considered to determine improvements in animal welfare. Although the test measures brief instances of positive behaviors, if proper testing procedures and measurements are considered, the results may more accurately reflect the animals’ positive affective states. It may be that in our study, increased duration and control of the intervention for all groups and extended post-cognitive bias tests would have better reinforced the effects of the toys and led to more accurate measures of cognitive bias.

### 4.3. Measuring Welfare: Appetite and Activity

Appetite is measured in many welfare studies; however, its connection is less than clear. In both human and non-human animals, positive and negative affect have been shown to moderate the rate of food consumption and food intake. However, the relationship between appetite and emotion is not simple. For instance, both negative feelings of stress or positive feelings of pleasantness may induce either increased or decreased food consumption, depending on the context and individual [56,57,58].

Activity is similarly equivocal. Increased exercise has been shown to reduce cortisol (commonly correlated with stress) levels in dogs in shelter environments [59]. For kenneled dogs, “activity” levels may include undesirable behaviors such as jumping when greeting or a number of transitions between activities, as well as vocalization or bodily postures [28,34]. It is not clear that the change in such behaviors, which is noted as an improvement in welfare for kenneled dogs, can be considered similarly in pet dogs.

Schipper et al. [35] found that providing feeding toys promoted appetitive behaviors and increased activity levels (time spent moving) in dogs. Our results did not align with these findings; After participating in the enrichment intervention, there were no significant differences in appetite in dogs who received toys. In both higher-level and lower-lower level subjects, a decrease in average duration of food consumption from pre- to post-intervention was observed (Figure 3). Though, it is important to note that we defined appetite as the time spent eating, whereas Schipper et al. considered appetitive behaviors as the time spent interacting with the food-based toy. This difference in measurement offers contrasting implications of what constitutes improved appetite. It may be that their results may reflect more on the behaviors of dogs housed in kenneled environments than on the efficacy of feeding enrichment on canine welfare. This is highlighted by Hubrecht [28], who found dogs who received enrichment spent significantly less time inactive—though subjects spent 51% of their time inactive before the intervention (quite different from companion dog populations). Additionally, Heys et al. [27] emphasized the human perception of toys’ effects on their dogs. Humans who use food-based toys were more likely to report more appetitive behaviors during feeding time in their dogs and less undesirable food-seeking behaviors such as begging. Dogs who received food enrichment toys were also less likely to exhibit issues with training and dog-directed fear. Whether increases in activity or appetite constitute positive measures of welfare is debatable—and highly dependent on subjects’ start state and environment.

### 4.4. Methodological Challenges

We found that our protocol inadvertently contained a few intrinsic issues that might help explain our lack of an effect of the intervention. For instance, the control group, while given intervention instructions matched to the experimental group for human engagement and food rewards, otherwise retained their normal routine. And per the survey responses, 67% of subjects in the control group were reported to have regular interactions with Level 2 or Level 3 toys, and all had Level 1 toys. Thus, there was not enough possible contrast between the experimental and control groups—especially after the mis-assignment of toys to the experimental group. Additionally, our sole measurement of activity levels was based on data from FitBark 2. Readings from accelerometers may provide more accurate measurements depending on the context of activity. In dogs, Colpoys and DeCock [60] found that data from FitBark 2 were more accurate when tracking off-leash activities (e.g., exploring a room) and less accurate in on-leash activities (e.g., walking with a person). Since we had no way of discerning the type of active behaviors in the subjects’ data, our measurements may not have provided the most accurate representation of dogs’ activity levels.

In the experimental groups, participants were instructed to engage with their dogs in their assigned toy condition for five minutes daily for 12 days between the two cognitive bias tests. This protocol of engagement may not have been sufficiently different from subjects’ normal lives to promote even short-term changes in behaviors. Other studies with even shorter durations (e.g., 3–5 days) of interactions with toys reported increased feeding and relaxation behaviors and decreased undesirable behaviors, but these used kenneled dogs [28,34,35,61].

By characterizing toys along levels in terms of sensory modalities and body parts engaged, we hoped to identify whether levels of complexity contributed to the toys’ efficacy in improving dogs’ welfare. Our unintended decrease in sample size, due to the mis-assignment of some toys, limited our ability to speak on this question. Notably, most of the dogs in our study were regularly given toys of at least Level 1 and Level 2 complexity, so the question of interest might not be the level of complexity of the toy, but the way that the dog (or dog and human) engages with it. It is likely that most dogs in the experimental group already had at least minimal prior exposure to one or more of their assigned toys or toy types. This may have led to habituation with the toys or potentially introduced unintentional aversive effects such as negative anticipation of the scheduled interactions with the toys. However, kenneled dogs have been shown to not habituate to various toys that we called Level 1 toys, even after two months of continued interactions [28].

Finally, it is worth reconsidering what constitutes “enrichment” for individual dogs. Effective enrichment must account for animals’ needs to express species-specific natural behaviors [56]; in addition, individual dogs may have idiosyncratic needs and preferences [28,29,30,35,61,62]. For instance, Pullen et al. [29] compared the effects of toys robust in texture and form (e.g., braided toy, durable toy) and less robust toys (e.g., plush toys with squeakers) and how they are oriented when provided to different populations of kenneled dogs. They found that subject dogs preferred toys presented on the floor (as opposed to being suspended) as well as less robust toys. Moreover, Hubrecht [28] observed the effects of different types of enrichment (conspecific socialization, human socialization, and toys) on kenneled dogs and found enriched dogs spent less time socializing and initiating play with conspecifics than dogs that did not receive enrichment. As the experimental groups of these studies received multiple enrichment conditions, the extent to which toy engagement contributed to those results was unclear. Still, these results suggest dogs may have preferences for particular toy types and engagements with their environment (e.g., socialization, object play).

The toys that we provided for our experimental group may not have been considered interesting to the subjects or not interesting enough to sustain interactions throughout the experimental timeline. In fact, Hunt et al. [61] found food-based enrichment induced the fewest behavioral changes in his subjects. We did not collect data on the intensity of engagement the subjects had with the toys throughout their sessions and therefore were unable to control for the effects of toy preference.

## 5. Conclusions

Our results add to the literature investigating whether dog toys provide enrichment of more than a momentary nature, effectively improving dog welfare. While our results were equivocal, they did point to the salience of the starting condition for subject populations. Subjects coming from a richly provisioned home may not see the same effects of adding a toy to their environment than kenneled or shelter-housed dogs would. Our study also highlights the challenges in using the most available methods of measuring welfare to look at changes in a companion dog population.

## Figures and Tables

**Figure 1 animals-13-03340-f001:**
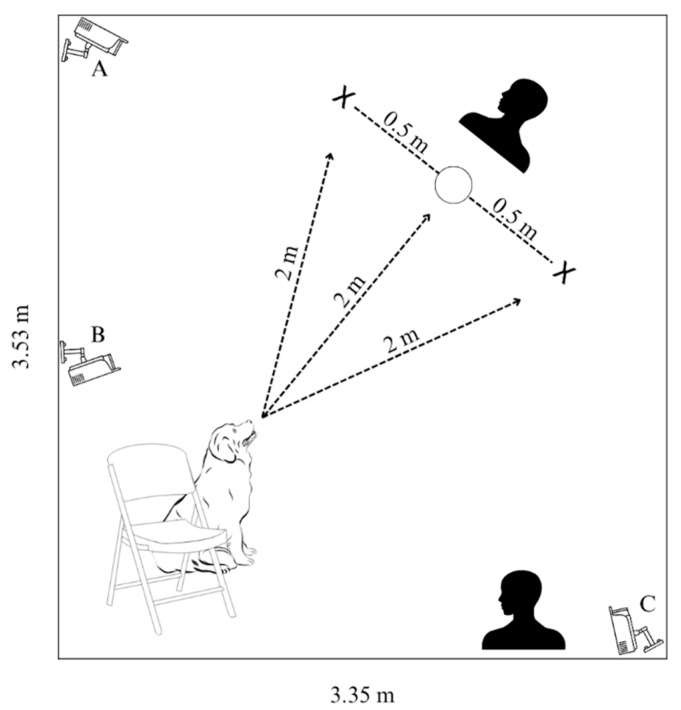
Cognitive bias test experimental stage; The dog is with their person at the starting position (chair) located 2 m away from the experimenter who presents the stimulus (bowl). The positive and negative bowls (L/R) are each 0.5 m away from the ambiguous bowl (center). A second experimenter to the side of the experimental area times subjects’ latencies to approach the bowls. Cameras A and C capture overhead views and camera B faces the stimuli from the starting position, at the shoulder height of a medium-sized dog.

**Figure 2 animals-13-03340-f002:**
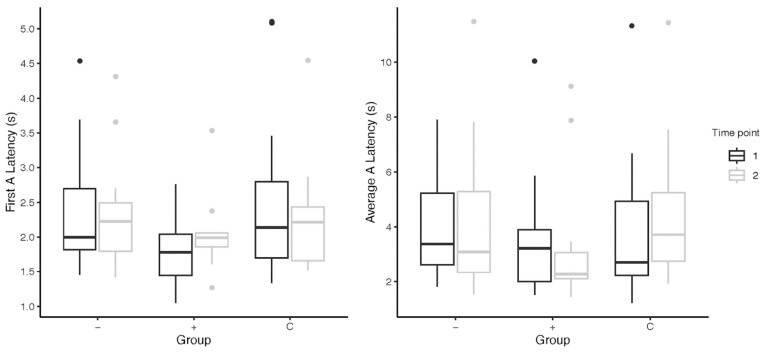
Subjects’ latency to approach the A-trial(s) from pre- to post-cognitive bias tests; “−” = lower-level subjects; “+” = higher-level subjects; “C” = control subjects. Points above the box plots indicate statistical outliers included in data analysis.

**Figure 3 animals-13-03340-f003:**
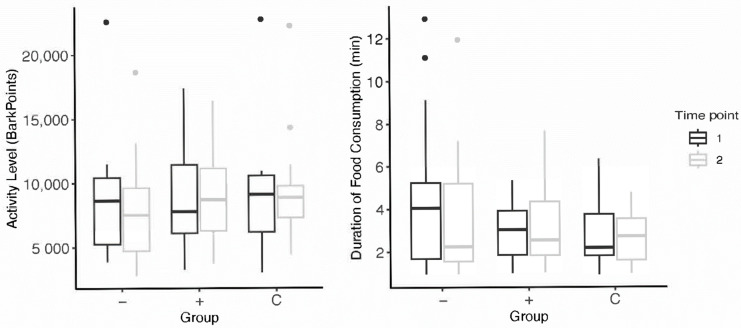
Subjects’ pre- and post-intervention data for activity levels and appetite; the “−” group refers to subjects who received lower-level toy assignments and the “+” group, higher-level toy assignments. Points above the box plots indicate statistical outliers included in data analysis.

**Table 1 animals-13-03340-t001:** Summary of demographic data of subject population.

**Average Age (Years)**				**Sex**			**Breed**	
<4	4–8	>8		M	F		Mixed	Purebred
53%	28%	19%		53%	47%		72%	28%
**Level of Favorite Toy**				**Group, Level from 1–3 (if applicable)**				
1	2	3	Other *	C		E, 1	E, 2	E, 3
79%	4%	11%	6%	32%		26%	21%	21%

Note: “E” = Experimental group; “C” = Control group; * “Other” refers to the subjects reported to have no favorite toy or who provided an ambiguous answer. See Appendix A for details on individual subjects.

**Table 2 animals-13-03340-t002:** Experimental timeline for each group.

Group	Timeline	Data Collected
	3 days	Start food consumption and activity data collection
Control	First visit	First cognitive bias test
	12 days *	Specified at-home activities
	Second visit	Stop food consumption and activity data collection Second cognitive bias test
	3 days	Start food consumption and activity data collection
Experimental (all levels)	First visit	First cognitive bias test
	12 days *	Specified at-home activities with new provided toy every 4 days
	Second visit	Stop food consumption and activity data collection Second cognitive bias test

* Subjects’ duration of data collection varied from 6 to 25 days in between each visit.

**Table 3 animals-13-03340-t003:** Provided toy level and toy characteristics.

Group	Provided Toys		Functions		
			Food Reward (Y/N)	Sensory Modalities	Engagement Behaviors
Control	-		Y		
Experimental, Level 1	Nylabone DuraChew Original Bone	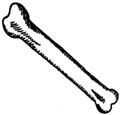	N	Touch	Chewing, pawing
	Max and Neo Furry Pals Water Bottle	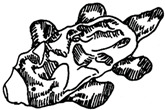			
	ChuckIt! Ultra Ball	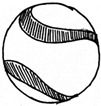			
Experimental, Level 2	LickiMat	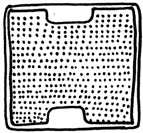	Y (visible)	Touch, smell, taste	Chewing, pawing, licking
	Kong Original	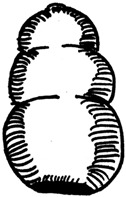			
	Starmark Treat Crunching Barbell	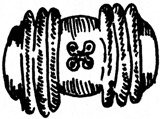			
Experimental, Level 3	Nina Ottosson puzzles:Treat Maze	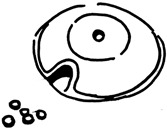	Y (invisible)	Touch, smell, taste	Chewing, pawing, licking, nosing, rooting, active investigation
	Brick	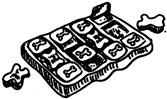			
	Tornado	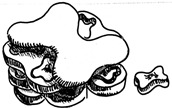			

**Table 4 animals-13-03340-t004:** Results of analyses on the effects of type of toy on subjects’ behavior.

Pre- to Post-Comparisons	Higher- vs. Lower-Level Toy Groups				Test Statistic	*p*	*rrb*
	**Higher**		**Lower**				
	*M*	*SD*	*M*	*SD*			
Cognitive bias: Avg A latencies	−0.23	1.44	0.24	2.61	93	0.905	0.03
Appetite	0.26	1.24	−0.66	1.94	−0.10	0.318	−0.24
Activity levels	−763.72	2500.29	−773.20	1687.34	97	0.592	−0.12

Note: Each comparison group consisted of different subject numbers due to omissions described in the text.

**Table 5 animals-13-03340-t005:** Results of analyses on the effects of toy provision on subjects’ behavior.

Pre- to Post-Comparisons	Higher-Level Toy vs. Control Groups				Test Statistic	*p*	*rrb*
	**Higher**		**Control**				
	*M*	*SD*	*M*	*SD*			
Cognitive bias: Avg A latencies	−0.23	1.44	0.51	3.71	109	0.373	0.21
Appetite	0.26	1.24	−0.22	1.57	−0.64	0.519	−0.16
Activity levels	−763.72	2500.29	203.73	1984.77	0.614	0.614	0.12

Note: Sample sizes ranged from 10 to 13 subjects for the higher-level toy group and from 13 to 15 subjects for the control group, as described in the text.

## Data Availability

The data presented in this study are available upon request from the corresponding author. The data are not publicly available due to privacy concerns and confidentiality of subjects’ identifying information.

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
