# Peer review of "Here Puppy, Chew on This: Short-Term Provision of Toys Does Not Improve Welfare in Companion Dogs"

_animals, 2023, doi:10.3390/ani13213340_

Round 1
Reviewer 1 Report
This study aims at investigating the effect on some behavioural measures of welfare of providing different types of toys as enrichment to family dogs.
The topic of this study is potentially interesting from an applied (dog welfare) perspective. However there are some methodological flows, which I am not sure the authors can solve.
My main concern is that the control group does not seem to be appropriate. While the dogs in the experimental group received brief daily interactions with different types of toys, in the presence of the owner, dogs in the control group received social interaction with their owners who was instructed to socialize with the dogs and provide food. I am concerned that this social interaction may be at least equally “enriching” for the dogs as the provided toys – especially considering the social predisposition of dogs and their flexibility in interacting with humans as social partners. In fact, no significant differences were found between experimental and control group.
Why did the authors expect that the provided toys may be more enriching (i.e., would provide more benefits to the welfare of dogs) than social interactions with the owner, who also provided treats?
I can also imagine that the owners in the control group, who were instructed to socialize with their dogs and provide treats, if there were no specific and detailed instructions on how to do this, could also have engaged in activities such as asking the dogs to perform some trained actions (e.g., sit, down etc.) to receive the treats. This may have provided a cognitive (and motor) activity for the dogs that I would say could be at least as enriching as using the toys given to the dogs in the experimental group.
Additionally, the authors report that “67% of subjects in the control group were reported to have regular interactions with Level 2 or Level 3 toys, and all have Level 1 toys”. This definitely makes these subjects a completely inappropriate control group.
In the light of these concerns, I am afraid I cannot recommend this manuscript for publication.
However, I found this approach of testing family dogs and discussion of the results in the light of results obtained on kenneled dogs, very useful and I would encourage the authors to find a way to solve those issues. If those can be solved, I would be happy to revise again this manuscript.
Author Response
Comment 1: I am concerned that this social interaction [for the control group] may be at least equally “enriching” for the dogs as the provided toys – especially considering the social predisposition of dogs and their flexibility in interacting with humans as social partners…Why did the authors expect that the provided toys may be more enriching (i.e., would provide more benefits to the welfare of dogs) than social interactions with the owner, who also provided treats?
Response 1: Thank you for your comments. We wanted to note that for the experimental groups, the humans were instructed to monitor their dogs’ enrichment sessions and, if needed, to “show” how the toy can be manipulated (without forcing engagement). Although human socialization can be viewed as a component in both the enrichment and control conditions, we did not choose to focus on the effects of different types of enrichment in dogs. Rather, we attempt to investigate whether findings of the effects of toys on kenneled dog populations can be generalized to companion dog populations (as it is often done).
We agree that this point was an important observation of the control group and may have been a contributing reason for why we did not see significant differences in behaviors between the groups. We explore the implications of this observation for our study and in future research in lines 587-592 and 596-597.
Comment 2: I can also imagine that the owners in the control group, who were instructed to socialize with their dogs and provide treats, if there were no specific and detailed instructions on how to do this, could also have engaged in activities such as asking the dogs to perform some trained actions (e.g., sit, down etc.) to receive the treats. This may have provided a cognitive (and motor) activity for the dogs that I would say could be at least as enriching as using the toys given to the dogs in the experimental group.
Response 2: We appreciate this observation. It is correct that our instructions for the “enrichment” sessions for the participants in the control group were flexible. Examples of the activities included simple obedience training and sitting by their dog while feeding treats for 5 minutes daily. While we acknowledge there is evidence that training (in general) may have a positive effect on cognitive, physical, and/or behavioral changes in dogs, in the context of our study’s procedures, we do not believe these activities extend beyond what the person would regularly do with their dog. Similarly, the same may apply to the experimental groups who also interacted with the toys for 5 minutes daily. We explore this as a potential reason for our findings in lines 477-480.
Still, we believe some interesting relationships between the groups are worth exploring. It may be that different activities (e.g. toys or human socialization) might point towards an engagement preference in dogs (lines 592-596) or that more longitudinal measures are needed to more accurately study the effects of toys on canine welfare (lines 505-511, 565-568).
Comment 3: Additionally, the authors report that “67% of subjects in the control group were reported to have regular interactions with Level 2 or Level 3 toys, and all have Level 1 toys”. This definitely makes these subjects a completely inappropriate control group.
Response 3: Expanding from Response 1, we agree that this was an interesting and potentially important observation of the control group. We reflected on our findings and the methodology used to study canine welfare in both our subject populations (lines 475-481). We believe it would be a challenge within the companion dog population to recruit subjects that have not interacted with basic toys similar to those we defined as Level 1 toys (e.g. plush toys, chewing bones). We attempted to ensure that we could, in fact, test whether toys were “enriching” by assigning subjects to a higher-level toy group and comparing their behaviors to both subjects who received lower/same-level toys and subjects who did not receive toys.
Comment 4: However, I found this approach of testing family dogs and discussion of the results in the light of results obtained on kenneled dogs, very useful and I would encourage the authors to find a way to solve those issues.
Response 4: We appreciate your comments and questions. We believe the observations on our control group are not a concern for the question we are asking in this study and that specific requirements for the control (and experimental) group(s) should be implemented in future studies. For instance, we could see considering the Level 1 group of subjects (or something similar involving a mix of toys we described as being Level 1 and 2) as the control group for future studies. However, as this study has not been conducted on companion dogs, we believe having a group of subjects that did not receive any toys was the appropriate measure for a control group.
Additional comments: We wanted to note that the lines we refer to in our responses were drawn from the first draft of the manuscript that you received.
Reviewer 2 Report
As providing a dog, e g when left alone, with a chewing toy is a frequent recommendation the study is highly relevant for improving dog welfare both as pets and in shelters.
Therefore I DO recoomend to publish the paper, however some improvements should be made:
a) literatuture: There are studies in humans and also I think lab animals on positive influences of chewing on dopamine/serotonin levels. These should be cited in the intro (sorry I cabnnot provide references, I am out of office currently
b) the reasoning that activity levels should be raised if welfare is improved ( e g by toys) is dubious to me, because dogs are described as lazy creatures by all researchers studying e g stray dogs
c) some weaknesses in the test situation (experimenter in the room, holder in the room and even know where the food is hidden) might impair the outcome (clever Hans!!) of the bias test and need to be discussed
d) statistics: as data are non-parametrical, medians, quartiles not means ( e g line 300, or SD line 325) should be used
why werde durations converted to minutes - line 334
which test exactly (line 380)?? Mann-Whitney U or Wilcoxon matched pairs??
As some data are repeatedly used a Bonferroni correction might be necessary
in data that are based on seconds, Randomization tests on interval level would provide higher resolution than ordinal tesrts
e) a few terms should be avoided : instinctual liebe 39, oweners several times) and the figs should be in the results, not discussion section
Author Response
Comment 1: There are studies in humans and also I think lab animals on positive influences of chewing on dopamine/serotonin levels. These should be cited in the intro
Response 1: Thank you for this suggestion. Although we could not find the papers you specifically mentioned, we added similar additions to the paragraph with lines 59-74.
Comment 2: the reasoning that activity levels should be raised if welfare is improved ( e g by toys) is dubious to me
Response 2: We acknowledge this comment and attempt to explore the implications of using activity levels as a measure of welfare in dogs in lines 523-528. We initially chose this measurement since it was used in several studies of canine welfare (Schipper et al., 2008; Herron 24 al., 2014; Hubrecht, 1993; Menor-Campos et al., 2011) and briefly explain the findings of two of these studies in lines 66-68 and 71-74. However, these studies were conducted on the kenneled dog population, which may consider activity in a different context than the companion dog population. In the kenneled dog population, increased activity can refer to different behaviors (lines 524-527) and is often considered a positive indicator of welfare. As there are no experimental studies that investigate the effects of toys on the companion dog population, we believe activity levels are an appropriate measurement of canine welfare for this study.
Comment 3: some weaknesses in the test situation (experimenter in the room, holder in the room and even know where the food is hidden) might impair the outcome (clever Hans!!) of the bias test and need to be discussed
Response 3: We acknowledge that there is potential for human-induced bias to influence the subjects’ in-test behavior. We also want to note that the procedures we followed for the cognitive bias test were extended from Mendl et al. (2010) and Duration & Horowitz (2019) and have been implemented across species (lines 99-100). We attempt to ensure the subjects’ know where the food is hidden by conducting numerous training trials (in both visits) to ensure they have learned the association between the bowl location and reward before moving on to the testing phase (see lines 208-211 for the criterion for “passing” the training phase).
Additionally, it is standard protocol with companion dog research to have a handler (blinded to the testing conditions)—which we had as the owner—and/or a researcher conducting the trials. We attempted to mitigate the effects of human influence on subjects’ in-test behaviors by providing strict instructions for the humans to not engage with their dogs (e.g. pointing) and researchers to not engage with the bowls or dogs (e.g. looking, vocalizing) during the trials. Despite our efforts, we acknowledge there are some challenges to the procedures for conducting the cognitive bias test and explore solutions that can be implemented in future studies in lines 506-511.
Comment 4: statistics: as data are non-parametrical, medians, quartiles not means ( e g line 300, or SD line 325) should be used
Response 4: Thank you for these suggestions. In line 300, we referred to comparing the average-A latencies of the subjects. However, we also chose to report the means (and standard deviation) in the results due to having used the averages as data. For this same reason, we choose to use standard deviation to determine the outliers of our data (referred to in line 325).
Comment 5: why were durations converted to minutes - line 334
Response 5: Owners were asked to report their dogs’ appetitive behavior (duration of eating) in minutes and seconds (mm:ss). A few subjects only reported their data in minutes. For this reason and for ease of interpreting the data in the results and figures, durations were converted to minutes and rounded to the hundredth of a minute.
Comment 6: which test exactly (line 380)?? Mann-Whitney U or Wilcoxon matched pairs??
Response 6: We apologize for the confusion this may have given you. The Wilcoxon signed ranks test was used to compare differences in P-empty and P-test trials from pre- to post-intervention in all subjects. All other comparisons were made using the Wilcoxon Mann-Whitney U-test. The test used in line 380 was the Wilcoxon Mann-Whitney U Test.
Comment 7: As some data are repeatedly used a Bonferroni correction might be necessary
Response 7: We appreciate this suggestion. We can make the correction if the editor finds it necessary, however we believe it is not since the groups are not being used for the same comparisons (as it is in multiple comparison tests). We interpreted the analyses on a per-test basis with comparisons.
Comment 8: in data that are based on seconds, Randomization tests on interval level would provide higher resolution than ordinal tesrts
Response 8: Thank you for this suggestion. We decided to keep our current statistical analyses since our data is nonparametric. The Wilcoxon and Kruskal-Wallis tests are appropriate for either ordinal or interval data and raw data were converted to ranks for analyses in these tests.
Comment 9: a few terms should be avoided : instinctual liebe 39, oweners several times)
Response 9: Thank you for bringing this to our attention. We have corrected these terms to more appropriate phrases.
Additional comments: We wanted to note that the lines we refer to in our responses were drawn from the first draft of the manuscript that you received.
Reviewer 3 Report
Overall, a new novel study that I was actually interested in reading. I look forward to reading more around this topic. But overall great study!
Introduction:
The introduction was very well written, with a good focus and foundation on the grasp of the area of study. The only feedback will be reducing the hypothesis are from 101 to 108, this will help allow the reads to get a condensed versus of the over hypothesis.
Methods:
Overall, the methods were very extensive and comprehensive, I would advise condensing this for readers. Some helpful areas to condense are below:
Line 121: 121 to 124 c can be reduced or reworded.
Line127-132: Can be removed if you are keeping table 1.
Table 1: possible placed in to appendix, and condensed into a general overview of Age, Sex, Breed, toys, and group descriptive stats. The Rows can be grouping.
Lines 141 to 150: can be reduced, give readers the basic overview of what was happening, lines 144 to 145 can be removed.
153 to 155 can be removed or reduced.
Section Training: line188
I would either reduced this to a condensed version and place the full version in appendix.
section 2.4
I would look to try to reduce this section a little for readers, and place more comprehensive information in appendix for individually wanting to dive into the fun aspects of the methods. I find it very interesting and novel, but if readers, if certain information is not relevant then we do not need to keep it. AKA activity levels. Especially as there is no significant information related to it in results. But I guess there is no relevant significance in a lot of the results.
Results:
I would actually consider rerunning the statistics using a different model. Without seeing the data, and the information that was fully collected, I would still consider a different approach. I would think that using a different statistical model would give more relevant information. Can you articulate the reasoning for the models chosen and why other models were not used?
Discussion
No comments amazing work!
Author Response
Comment 1: The only feedback will be reducing the hypothesis are from 101 to 108, this will help allow the reads to get a condensed versus of the over hypothesis.
Response 1: Thank you for this suggestion. We agree that it was redundant and condensed the hypotheses for each measure of welfare into one sentence.
Comment 2: Overall, the methods were very extensive and comprehensive, I would advise condensing this for readers.
Response 2: We greatly appreciate these comments and followed your suggestions (more below).
Comment 3: Line 121: 121 to 124 c can be reduced or reworded.
Response 3: We reduced the explanations of our criteria during subject recruitment to a few variables we thought were interesting or important to include.
Comment 4: Line127-132: Can be removed if you are keeping table 1. Table 1: possible placed in to appendix, and condensed into a general overview of Age, Sex, Breed, toys, and group descriptive stats. The rows can be grouping.
Response 4: We summarized the table of subject demographics into a condensed table and moved the original table to the Supplementary Materials. For this reason, we wished to keep the long-form description of the subject breeds.
Comment 5: Lines 141 to 150: can be reduced, give readers the basic overview of what was happening, lines 144 to 145 can be removed. 153 to 155 can be removed or reduced.
Response 5: We agreed that the explanations of on-boarding subjects prior to their in-person visits were extensive and can be confusing when presented all at once. We reduced explanations of the experimental procedure and instructions given to the humans about data collection of their at-home activities. We reduced details about the procedures of the cognitive bias test and moved additional information to the Supplementary Materials.
Comment 6: line188 I would either reduced this to a condensed version and place the full version in appendix.
Response 6: We reduced explanations of the procedures of training and testing trials for the cognitive bias test and moved details we deemed interesting or important to the Supplementary Materials.
Comment 7: I would look to try to reduce this section a little for readers, and place more comprehensive information in appendix for individually wanting to dive into the fun aspects of the methods. I find it very interesting and novel, but if readers, if certain information is not relevant then we do not need to keep it. AKA activity levels. Especially as there is no significant information related to it in the results. But I guess there is no relevant significance in a lot of the results.
Response 7: We appreciate this suggestion. We reduced procedures for the behavioral analyses of the cognitive bias test. Specifically, we moved details of adjusted average A-latencies and the first A-trials to the Supplementary Materials and briefly summarized these procedures in the main paper.
Additionally, we acknowledge and appreciate your comment on excluding less relevant details such as the activity levels. While we consider comparisons of measurements of welfare separately and acknowledge that the primary results refer to the cognitive bias test, we wish to keep the analyses and results pertaining to measures of appetite and activity levels in the main paper. We feel this is appropriate since our original research question attempts to investigate whether findings of the effects of toys on kenneled dog populations can be generalized to companion dog populations and appetite and activity levels are commonly explored in such studies.
Comment 8: I would actually consider rerunning the statistics using a different model. Without seeing the data, and the information that was fully collected, I would still consider a different approach. I would think that using a different statistical model would give more relevant information. Can you articulate the reasoning for the models chosen and why other models were not used?
Response 8: Thank you for this suggestion. Expanding on the last sentence of our 7th response, since the effects of toys on (multiple) measures of welfare have not been studied (using experimental methods) in companion dogs, we believe a simpler statistical model would be most appropriate in investigating our question. We agree that more complicated regression models that are able to consider fixed and random effects such as age, gender, or prior experience/engagement with certain types of toys yield more interactions between variables. Similarly, we acknowledge that improvements in methodology (such as to those challenges we mentioned in lines 505-511, 527-528, 562-564, and 597-599) would yield more complex and accurate data—translating to more complex analyses and results—for future studies.
However, given 1) the limitations of that data type, 2) smaller sample sizes between control, experimental, and sub-experimental groups (and even smaller sample sizes due to the misassignment of toy levels), and 3) our intentions of considering each measure separately, we believe our statistical model is appropriate and wish to keep them the same.
Additional comments: We wanted to note that the lines we refer to in our responses were drawn from the first draft of the manuscript that you received.